# No Filter: Cultural and Socioeconomic Diversity in Contrastive Vision–Language Models

**Angéline Pouget**[*†]
apouget@ethz.ch

**Lucas Beyer**
lbeyer@google.com

**Emanuele Bugliarello**
bugliarello@google.com

**Xiao Wang**
wangxiao@google.com

**Andreas Peter Steiner**
andstein@google.com

**Xiaohua Zhai**
xzhai@google.com

**Ibrahim Alabdulmohsin**[†]
ibomohsin@google.com

Google DeepMind

## Abstract

We study cultural and socioeconomic diversity in contrastive vision–language models (VLMs). Using a broad range of benchmark datasets and evaluation metrics, we bring to attention several important findings. First, the common filtering of training data to English image–text pairs disadvantages communities of lower socioeconomic status and negatively impacts cultural understanding. Notably, this performance gap is not captured by—and even at odds with—the currently popular evaluation metrics derived from the Western-centric ImageNet and COCO datasets. Second, pretraining with global, unfiltered data before fine-tuning on English content can improve cultural understanding *without* sacrificing performance on said popular benchmarks. Third, we introduce the task of geo-localization as a novel evaluation metric to assess cultural diversity in VLMs. Our work underscores the value of using diverse data to create more inclusive multimodal systems and lays the groundwork for developing VLMs that better represent global perspectives.

## 1 Introduction

Contrastive vision–language models (VLMs) have emerged as a powerful and versatile method to bridge the gap between visual and textual information in deep learning systems. They utilize a dual-encoder architecture to map both images and texts into a shared latent space. Representations in this latent space are learned leveraging large datasets of noisy image-text pairs from the web. Work including CLIP [46], ALIGN [31] and SigLIP [75] validates this approach at scale with impressive zero-shot transfer results across a wide range of downstream tasks.

However, due to the growing range of applications for contrastive VLMs, it is imperative to evaluate them not only with respect to standard performance metrics, such as their classification accuracy on ImageNet-ILSRCV2012 [17] or image-text retrieval performance on COCO [38], both of which are Western-oriented [53, 16], but also in terms of "cultural diversity." We illustrate what we mean in Figure 1 (a) and (b). When performing zero-shot classification on images from the Google Landmarks Dataset (GLDv2) [66], SigLIP models trained on only English-language image–text pairs (henceforth

---

[*]Work performed while interning at Google DeepMind.
[†]Corresponding authors.

38th Conference on Neural Information Processing Systems (NeurIPS 2024).

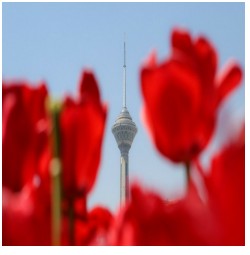 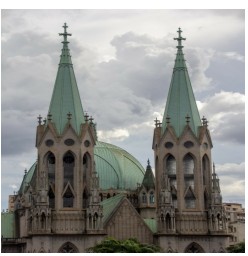 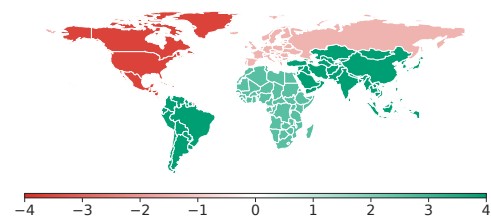

(a) **en**: CN Tower (Canada)
**globe**: Milad Tower (Iran)

(b) **en**: Notre-Dame Basilica of Montreal (Canada)
**globe**: Catedral da Sé de São Paulo (Brazil)

(c) 0-shot accuracy improvement [%] on the Dollar Street dataset when switching from English-only (**en**) to global data with English translation (**globe-tl**).

Figure 1: Models trained on English image-text pairs exhibit a lack of diversity when evaluated on images from other regions, sometimes confusing landmarks with similar ones located in the West.

denoted **en**) tend to misclassify international landmarks as similar-looking landmarks located in English-speaking countries. Note that this is the currently prevalent pretraining approach in the literature. In contrast, SigLIP models trained on the full, global data (henceforth denoted **globe**) identify the correct landmarks.

A similar observation can also be made in Figure 1 (c), where we compare the zero-shot classification accuracy of a model trained on **en** data to that of a model trained on **globe** data with its text translated to English using the Google Translate API (henceforth denoted **globe-tl**). As can be seen, the **en** model seems to be biased towards data from Western regions. Switching to **globe-tl** data lowers performance for images from North America and Europe but significantly improves performance for other regions such as South America and Africa that are traditionally underrepresented in AI.

It is worth clarifying that cultural diversity in our context is different from "fairness" and the mitigation of societal stereotypes. While recent work has shown that CLIP models perpetuate and amplify social biases and stereotypes present in the training data [27, 7, 2, 43], our emphasis is different. We focus on improving the ability for a VLM to recognize and accurately interpret visual and textual data from a wide range of geographical, socioeconomic and cultural contexts, such as physical surroundings, traditions, customs and everyday goods. Evaluating cultural diversity is critical because it ensures that VLMs perform well across diverse environments and that they recognize and respect the varied perspectives that exist worldwide.

In this work, we present a comprehensive study of cultural and socioeconomic diversity, focusing on the impact of training data source, composition and processing (including translation), using the recently introduced SigLIP models [75] as a case study. We cover a wide range of benchmark datasets and evaluation metrics. Amongst our findings, we show that while SigLIP models trained on English-only image–text pairs (**en**) achieve state of the art results on popular benchmarks (ImageNet, COCO), this filtering disproportionately hurts model performance for low-income households and regions, and negatively impacts cultural diversity. Crucially, these **en** models achieve demonstrably lower performance on cultural diversity benchmarks *even after fine-tuning on more diverse and global data*. Conversely, models pretrained on the full, global data (**globe**, **globe-tl**) followed by brief English-only fine-tuning can match and even outperform the English-only baselines on the popular Western-oriented benchmarks, while also performing demonstrably better in cultural diversity benchmarks. Hence, pretraining on global data yields better foundation models.

In addition, we introduce the task of geo-localization—based on datasets such as XM3600 [58], Dollar Street [49], GeoDE [47] and GLDv2 [66]—as a novel evaluation metric for cultural diversity. We show that, unlike, for example, XM3600 retrieval that evaluates multilinguality, geo-localization is strongly dependent on the global composition of the dataset used during pretraining.

In line with Alabdulmohsin et al. [2], we focus on contrastive learning for several reasons. These models have a wide range of applications, e.g. zero-shot classification and cross-modal retrieval, and are being increasingly adopted in critical domains like healthcare [76, 52], and as a backbone for other models [73, 72, 41]. We study SigLIP [75], the currently best-performing and a widely used CLIP-

style model. SigLIP and CLIP operate on the same principle of aligning representations/embeddings for texts and images and the difference is only in the choice of the loss function.

## 2 Preliminaries

**SigLIP Overview.** Given a mini-batch $\mathcal{B} = \{(I_1, T_1), (I_2, T_2), \dots\}$ of image–text pairs, an image encoder $f(\cdot)$ and a text encoder $g(\cdot)$, SigLIP aims to align the image embeddings $\mathbf{x}_i = f(I_i)/\|f(I_i)\|_2$ in the given batch with their corresponding text embeddings $\mathbf{y}_i = g(T_i)/\|g(T_i)\|_2$. The sigmoid-based loss $L(\mathcal{B})$ processes every image–text pair independently with a positive label $z_{ii} = 1$ for the matching pairs $(I_i, T_i)$ and a negative label $z_{ij} = -1$ for all other pairs $(I_i, T_{i \neq j})$:

$$L(\mathcal{B}) = \frac{1}{|\mathcal{B}|} \sum_{i=1}^{|\mathcal{B}|} \sum_{j=1}^{|\mathcal{B}|} \log\left(1 + \exp\left(z_{ij}(-t\mathbf{x}_i \cdot \mathbf{y}_j + b)\right)\right), \tag{1}$$

where $t > 0$ and $b$ are learnable temperature and bias parameters. We follow Zhai et al. [75] in most aspects: we use a Vision Transformer (ViT) [18] for images and a Transformer [60] for text, both of size Base (B), with an embedding dimension of 768. Images are resized to $256 \times 256$ resolution and we use $16 \times 16$ patch size. Text input is tokenized using a model trained on mC4 [69], a Common Crawl-based dataset covering 101 languages, with a vocabulary size of 250 k. We keep a maximum of 64 tokens. We use the modified Adafactor optimizer [54, 74] for all our experiments with an initial learning rate of $10^{-3}$, reverse square root decay, weight decay of $10^{-4}$ and 50 k warmup and cooldown steps. Training batch size is 16 k. Models are developed in the Big Vision codebase [6] using Tensor Processing Units (TPUs) [32]. Each model is trained on 10 B image–text pairs (roughly 610 k steps) for about 40 k TPUv2 core-hours, so they are compared on a *compute-matched* regime.

**Pretraining Data.** We base our analysis on a range of models trained on different subsets of the WebLI dataset, a high-volume image–text dataset collected from the public web [15]. Each example in our filtered subsets contains a caption in the original language as well as an English translation if the caption is not in English. Hence, we can distinguish between three different dataset variants: (1) **globe**, (2) **en**, and (3) **globe-tl**. Here, **globe** denotes the raw, *multilingual* data with minimal filtering applied (e.g., removing sensitive and personally identifiable information [15]). We denote its subset that contains only English captions by **en**. This mirrors the filtering that is currently applied in several influential papers, including CLIP, ALIGN and SigLIP, as well as the common way [22, 21, 67, 56, 57] of using LAION [51] and DataComp [22]. The third and last variant is **globe-tl**, which consists of the same images as **globe** but with an added pre-processing step in which any non-English text is *machine-translated* to English. We use this variant to differentiate between the impact of multilinguality, such as low-resource languages being severely underrepresented [8], and cultural diversity of the trained models. Since it is plausible that VLMs may not leverage their full multilingual potential when prompted in non-English languages, as has been observed for LLMs [20], we report results for both **globe** and **globe-tl** in all experiments and highlight any notable differences. To determine statistical significance of our results, we train 3 models each for **en**, **globe** and **globe-tl** with different random seeds. We perform two-sample t-tests and report the 95% confidence intervals.

**Evaluation Data.** To evaluate cultural diversity, we use five datasets: Dollar Street [49], GeoDE [47], GLDv2 [66], XM3600 [58] and MaRVL [39]. These datasets satisfy the criteria of being of sufficiently high quality and collected with geographical diversity in mind, see Figure 2. In addition, they can readily be used for evaluating contrastive VLMs without a decoder, by supporting either zero- or few-shot classification or cross-modal retrieval.

The Dollar Street (DS) [49] dataset encompasses 38 479 images depicting 289 household items commonly found in everyday settings across 63 countries. Each image is tagged with object descriptors and demographic data such as region, country, and monthly income. Following Rojas et al. [49], we map 96 topics to ImageNet classes, resulting in a subset of 21 536 images. This subset is split into training and testing sets (17 228 and 4 307 images, respectively). Notably, object tags (including the fuzzy matching to ImageNet labels) are not mutually exclusive. We use this dataset for zero-shot object classification and geo-localization. DS has been released under the CC BY-SA 4.0 license.

We also evaluate our models on a geographically diverse subset of the Google Landmarks Dataset v2 (GLDv2) [66], featuring 1 542 images representing 884 landmarks from 84 countries [35]. These

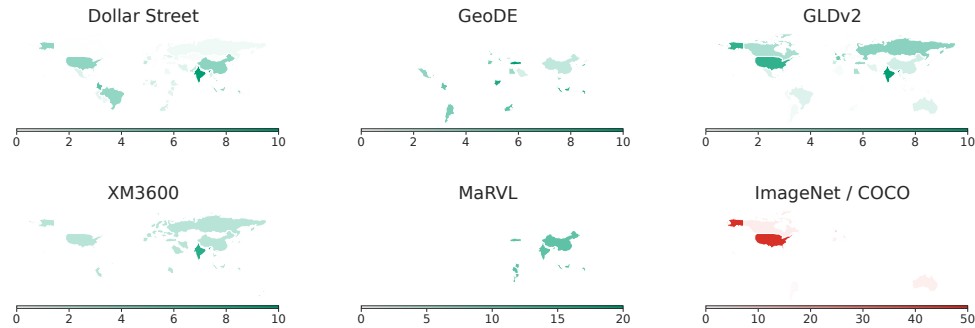

Figure 2: Data distribution [%] for each of the evaluation datasets, only approximate in MaRVL [39] based on the 5 languages collected in the dataset. Dollar Street [49], GeoDE [47], GLDv2 [66] and XM3600 [20] are geographically diverse. MaRVL is included because it focuses on underrepresented regions, such as Asia and East Africa. By comparison, ImageNet examples are mostly from a few Western countries (see for instance [53]). COCO has a nearly identical distribution to ImageNet [16].

images serve as the public and private test datasets for the 2021 Google Landmark Retrieval Challenge, providing a benchmark for evaluating landmark recognition algorithms [34]. We use it for zero-shot landmark classification using the English names of all 884 landmarks as class labels. GLDv2 images have CC-0 or Public Domain licenses. Annotations are licensed by Google LLC under CC BY 4.0.

GeoDE [47] is a geographically diverse dataset comprising 61 940 manually annotated images categorized into 40 classes. The dataset emphasizes object classes across six world regions: Europe, Americas, West Asia, East Asia, Africa, and Southeast Asia. Images were collected through crowd-sourcing, with rigorous manual verification procedures implemented to ensure data integrity. We use GeoDE for zero-shot object classification as well as for geo-localization. Since GeoDE does not have train and test splits, we shuffle the data and take the first 20 000 images for training and the rest for testing in the few-shot geo-localization evaluations. The dataset is licensed under CC BY 4.0.

The Multicultural Reasoning over Vision and Language (MaRVL) dataset [39] presents an ImageNet-style concept hierarchy designed to encompass a broader linguistic and cultural spectrum. The dataset incorporates five languages: Indonesian, Mandarin Chinese, Swahili, Tamil, and Turkish. Both the conceptual categories and associated images are curated exclusively by native speakers of each respective language. While the primary task on MaRVL involves validating statements concerning image pairs, the dataset holds potential for broader applications, including single-image evaluation metrics. The MaRVL texts and features are distributed under the CC BY 4.0 license. Image access is provided only for (non-commercial) research purposes.

Finally, Crossmodal-3600 (XM3600) [58] is a multilingual evaluation dataset that comprises 3 600 images accompanied by 261 375 human-generated reference captions spanning 36 languages. The dataset is sourced from the Open Images Dataset [36], with 100 images per language. Quality assurance measures, including a post-annotation verification process, attest to the overall high quality of the captions. We report both image-caption retrieval and geo-localization results. In the few-shot geo-localization evaluation, since XM3600 does not have train and test splits, we randomly shuffle the data and use the first 1 800 images for training and the second 1 800 images for testing. The annotations are licensed under the CC BY 4.0 license.

**Summary of Findings.**    Before digging into the detailed results, we summarize our key findings:

1. The currently predominant paradigm of directly or indirectly filtering the training data to English image–text pairs negatively impacts cultural diversity and disproportionately hurts communities of lower socioeconomic status, exacerbating existing disparities. Its impact is demonstrably captured by zero-shot classification accuracy on Dollar Street, GLDv2, GeoDE, and MaRVL (Section 3.1, Table 1, Figure 3). This has been known for vision datasets, such as ImageNet and OpenImages [53], but is relatively less explored in image-text pretraining data scraped from the Web.

2. The image features learned by models trained on such filtered data are less culturally diverse. We demonstrate and quantify this by introducing the few-shot geo-localization task. A

Table 1: Filtering training data to English image–text pairs negatively impacts cultural diversity but improves performance on standard benchmarks. Asterisk (⋆) denotes statistical significance at the 95% confidence level. No statistically significant differences are observed for XM3600 retrieval.

| | **en** | **globe** | **globe-tl** | **en** vs. **globe-tl** |
|---|---|---|---|---|
| **Culturally diverse zero-shot evaluations** | | | | |
| Dollar Street | 48.52 ±0.53% | 48.82 ±0.34% | 49.96 ±0.71% | +1.44%⋆ |
| GLDv2 | 43.84 ±0.52% | 46.18 ±1.30% | 49.46 ±1.17% | +5.62%⋆ |
| GeoDE | 91.82 ±0.39% | 92.00 ±0.10% | 92.84 ±0.05% | +1.02%⋆ |
| MaRVL Concepts | 68.30 ±0.50% | 69.09 ±0.28% | 69.96 ±0.35% | +1.66%⋆ |
| **Crossmodal-3600 (XM3600) retrieval top-1 recall** | | | | |
| *English captions* | | | | |
| Image → Text | 50.60 ±1.54% | 49.10 ±0.28% | 49.74 ±1.03% | −0.86% |
| Text → Image | 47.74 ±2.23% | 45.01 ±0.21% | 44.48 ±1.83% | −3.26% |
| *Native captions translated to English* | | | | |
| Image → Text | 62.73 ±1.28% | 60.70 ±1.38% | 62.02 ±1.41% | −0.71% |
| Text → Image | 56.49 ±1.28% | 52.60 ±1.21% | 54.13 ±1.28% | −2.36% |
| **Prevalent Western-oriented benchmarks** | | | | |
| 0-shot ImageNet | 70.36 ±0.28% | 66.81 ±0.18% | 68.23 ±0.19% | −2.13%⋆ |
| COCO I→T R@1 | 59.28 ±0.90% | 55.81 ±0.59% | 54.00 ±1.39% | −5.28%⋆ |
| COCO T→I R@1 | 42.91 ±0.56% | 38.09 ±0.58% | 37.78 ±0.23% | −5.13%⋆ |

linear probe on the image encoder shows **en** trained image encoders to be significantly less "world-knowledgeable" than **globe** or **globe-tl** trained image encoders (Section 3.2, Table 2).

3. These performance disparities are not reflected by, and even at odds with, the currently most popular (and often sole reported) benchmarks based on ImageNet [17] and COCO [38]. In addition, we present evidence that benchmarks used to evaluate multilinguality such as XM3600 [58], are *insufficient* to evaluate models' cultural diversity (Section 3.3, Table 1).

4. As a potential way out of this conundrum, we find that pretraining on unfiltered (global) data followed by fine-tuning on English-only data improves cultural diversity *without* sacrificing performance on the popular Western-centric benchmarks. This allows practitioners and researchers to strike a balance between these otherwise competing metrics. These performance improvements across benchmarks are further enhanced by translating training data to English (Section 3.4, Figure 4, Figure 5).

In summary, we call for a stop of pretraining on directly or indirectly English-filtered data.

## 3 Detailed Results

### 3.1 No filter for improved cultural diversity

To assess models' cultural diversity, we report zero-shot classification accuracy on Dollar Street, GLDv2, GeoDE and MaRVL. This evaluation includes tasks such as recognizing common household items across different countries and income brackets (Dollar Street and GeoDE), identifying significant landmarks and places of worship (GLDv2) and categorizing image concepts (MaRVL). Detailed results are provided in Table 1. Pretraining on globally diverse data yields substantial enhancements across all zero-shot classification metrics related to cultural diversity. However, these improvements stand in contrast to the popular benchmarks of ImageNet zero-shot accuracy and COCO retrieval scores. Given their prominence, it is not surprising that filtering training data to English image–text pairs, directly or indirectly, has quickly established itself as the preferred choice [46, 31, 75, 22, 21, 67, 56, 57]. When considering the cultural and geographical diversity of the resulting models however, a very different picture presents itself with models trained on **globe-tl** and **globe** outperforming those trained on **en** across all four benchmarks by a significant margin.

Indeed, a more fine-grained analysis of zero-shot classification on Dollar Street confirms that filtering to English-only training data disproportionately hurts low-income and non-Western communities;

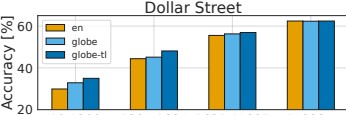 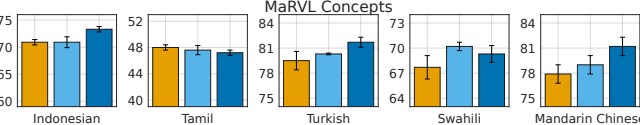

Figure 3: Filtering to English-only data further exacerbates existing performance disparities across socioeconomic subgroups. LEFT: Zero-shot classification results for Dollar Street, disaggregated by income level ($x$-axis). The performance difference between **en** and **globe-tl** is larger for lower-income households. Also, the performance disparity between the lowest and highest income groups is 32.5% in **en** (from 29.9% in $0-200 income group to 62.4% in $1998+ income group), but this gap is reduced (improved) to 27.4% in **globe-tl**. RIGHT: MaRVL Concepts classification accuracy disaggregated by each of the five languages/regions: Pretraining on **globe-tl** improves performance for Indonesian, Turkish and Mandarin Chinese and yields a similar performance to **en** for Tamil and Swahili.

see Figure 1 (right) and Figure 3 (left). In MaRVL, performance for all regions (except TA) tend to benefit from using globally diverse training data, as shown in Figure 3 (right). The difference in classification accuracy for different income groups and geographic regions is a worrying signal of the inherent biases of filtering to English-only data. Removing the English language filter unsurprisingly leads to a drop in performance for images from Western countries, as exemplified by ImageNet and COCO benchmarks, but it significantly improves performance for the rest of the world.

## 3.2 Few-shot geo-localization

The disparities between models' cultural diversity become even more pronounced when considering the image encoder's few-shot geo-localization performance, which we introduce in this work. This task involves learning to predict the geographical origin of an image, be it at the country or regional level, with only a limited number of training samples per location. We do this using a linear classification probe on the image representation, employing squared loss and L2 regularization – a problem that admits a closed-form solution. Our training dataset comprises a constrained number of images per location, with results reported for sample sizes of 5, 10, or 25 instances. Should the available examples for a given location in the training set fall below this threshold, we utilize all accessible samples. Our results (Table 2) suggest that few-shot geo-localization holds promise as a novel metric to assess cultural diversity in VLMs.

Independent of the prediction target being more fine-grained (country) or more coarse-grained (region), both **globe** and **globe-tl** have a significant edge over **en**, as shown in Table 2. These results suggest that models that are not trained on sufficiently diverse and global data fail to learn features that capture country- or region-specific information. Another point that is noteworthy, especially when comparing Table 1 to Table 2, is that the difference in performance between the **globe** and

Table 2: Global data improves few-shot geo-localization performance significantly. Performance differences are statistically significant for all reported results at 95% confidence level (*).

| Task | Shots | en | globe | globe-tl | en vs. globe-tl |
|---|---|---|---|---|---|
| Dollar Street (country) | 5 | 11.33 ±0.22% | 14.16 ±0.23% | 12.81 ±0.60% | +1.48%* |
| | 10 | 17.45 ±0.33% | 21.51 ±0.56% | 20.42 ±0.61% | +2.97%* |
| | 25 | 24.40 ±0.38% | 30.11 ±0.50% | 29.17 ±0.27% | +4.77%* |
| XM3600 (country) | 5 | 14.35 ±0.36% | 19.04 ±0.52% | 19.24 ±0.46% | +4.89%* |
| | 10 | 18.56 ±0.17% | 25.83 ±0.50% | 25.98 ±0.94% | +7.42%* |
| | 25 | 25.96 ±0.53% | 34.76 ±0.90% | 33.85 ±0.28% | +7.89%* |
| GeoDE (country) | 5 | 12.66 ±0.37% | 19.59 ±0.63% | 19.91 ±1.37% | +7.25%* |
| | 10 | 16.41 ±0.56% | 26.29 ±0.38% | 26.23 ±0.81% | +9.82%* |
| | 25 | 23.24 ±0.26% | 37.13 ±0.51% | 36.54 ±0.27% | +13.30%* |
| GeoDE (region) | 5 | 28.18 ±1.22% | 33.32 ±0.37% | 34.51 ±1.90% | +6.33%* |
| | 10 | 32.03 ±1.01% | 40.48 ±0.34% | 41.09 ±1.53% | +9.06%* |
| | 25 | 38.86 ±0.75% | 49.61 ±0.73% | 50.38 ±1.36% | +11.53%* |

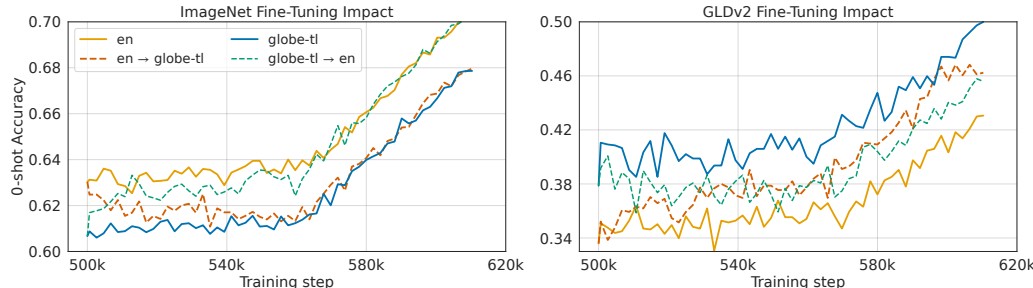

Figure 4: Fine-tuning **globe-tl** on **en** quickly catches up with **en** for ImageNet zero-shot evaluation while also performing better on GLDv2. Conversely, fine-tuning **en** on **globe-tl** does not suffice to close the gap in performance on culturally diverse benchmarks.

**globe-tl** models is notably smaller in the few-shot setting. This is not surprising since the standard zero-shot classification prompt templates and object or landmark names are all in English and, hence, this more closely resembles the **globe-tl** training data version and does not leverage the multilingual capabilities of the **globe** model. In the few-shot setting, by contrast, we only use the image encoder's representations and hence the impact of the text tower on the evaluation results is reduced.

### 3.3 Decoupling multilinguality and cultural diversity

As shown in Table 1, models trained on English-only data (**en**) perform best on Western-oriented benchmarks. New benchmarks such as XM3600 have recently been introduced to evaluate multilinguality in VLMs. To recall, XM3600 contains 100 images each from 36 different linguistic regions, captioned by native speakers in all 36 languages. At first, it might seem that performing image–text retrieval based on the English captions or the English translation of the native captions for all 3 600 images could serve as a viable signal for cultural diversity.

However, when comparing our models' performance on these two tasks, we do not find any statistically significant differences between the three models. Unsurprisingly, the **globe** model performs best when performing retrieval on *non-English* captions since the other two models have only seen English texts during training. However, when evaluating all models on the English-language captions or the English translations of captions in other languages, there are no statistically significant differences between the three model variants. We hypothesize that this is because XM3600 is derived from the Open Images dataset, which contains primarily Western images [53] or, images quite similar to what is already available in English domains. Closer inspection of the XM3600 images confirms that most images from non-Western countries are likely taken by tourists. For example, among the 100 images from the Arab world, 12% are images of cars. These images do not adequately reflect cultural differences. Moreover, since the original caption language is often English, similar images are likely to have been included in the **en** training data, explaining why there are no statistically significant differences between the three models. When comparing retrieval scores between English captions and English translations of captions in other languages across all three models, we observe a significant difference. Upon further investigation, we found that non-English captions tend to provide more detailed descriptions of the target image, leading to higher retrieval accuracy. This discrepancy might not be associated with cultural diversity or multilinguality but rather reflects variance in annotators.

Based on these findings, we argue that datasets originally created for evaluating multilinguality, such as XM3600, might not be sufficient for evaluating cultural diversity in multimodal systems.

### 3.4 Bridging the gap

**Fine-tuning.** As shown in Table 1, improving diversity generally results in a loss of performance on standard benchmarks. In general, we found that the two objectives typically *compete* with each other: in Figure 4, where we take two models pretrained on either **en** or **globe-tl** data and fine-tune them on the other dataset for a short duration. Clearly, improving culturally diverse metrics is accompanied by a loss in performance on Western-oriented benchmarks and vice versa. This is even more clear when looking at the correlation coefficients across metrics of over 40 models, shown in Figure 5 (b).

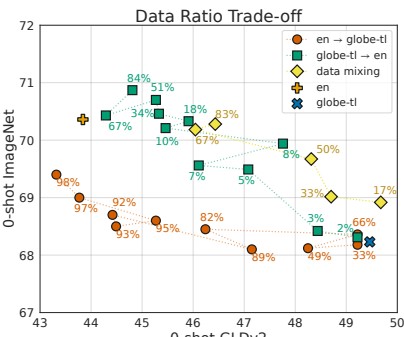 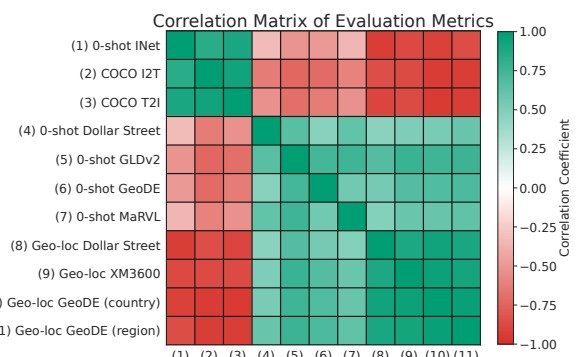

Figure 5: LEFT: Fine-tuning allows for a controlled trade-off between cultural diversity and performance on standard benchmarks. Fine-tuning **globe-tl** on **en** is strictly better than fine-tuning **en** on **globe-tl**, but mixing training data in different proportions achieves a better trade-off overall. Values in percentages [%] correspond to the fraction of time training is restricted to **en** data. RIGHT: Correlation coefficients of the evaluation metrics computed based on over 40 fully trained models.

In Figure 5 (a), this trade-off is further visualized. We first pretrain models on either **en** or **globe-tl** for a set number of steps (varying between $100k$ and $600k$) and we then switch to the other dataset for the remainder of the training duration. Each model is trained for approximately $610k$ steps in total. We plot the resulting performance at the end of training for each of these switch-over points. In line with what we see in Figure 4, we find that very little fine-tuning is sufficient to significantly impact model performance and improve performance either for standard benchmarks (green squares) or for the cultural diversity evaluations (red circles), depending on which dataset was used at the end. Hence, if we fine-tune for more than $100k$ steps, final model performance is mostly determined by the dataset that the model is fine-tuned on.

Notably, it is possible to improve performance on cultural understanding (e.g. improving 0-shot accuracy in GLDv2 from 44% in **en** to over 45.5%) *without* impacting ImageNet 0-shot accuracy by pretraining on **globe-tl** and fine-tuning that model on **en** data. However, the opposite does not hold. English-only pretrained models achieve lower performance on culturally diverse benchmarks even after fine-tuning on global data with the tradeoff curve being clearly suboptimal to the one observed when fine-tuning culturally diverse models on **en** data. Therefore, pretraining on globally diverse data before fine-tuning on English-only image–text pairs allows for a nuanced trade-off between catering to cultural diversity and achieving strong performance on well-established benchmarks.

**Data mixing.** Besides fine-tuning, we also study the impact of mixing the two versions of data during pretraining. Because **en** is a subset of **globe-tl**, mixing is equivalent to assigning more weight to English data. Figure 5 highlights that data mixing is as good as fine-tuning (if not better) in achieving a balance between Western-oriented and culturally diverse benchmarks. Moreover, it is applicable in settings where we may have more than 2 different data splits. However, data mixing entails training new models *ab initio*, thus incurring higher computational cost compared to fine-tuning. In our setting, we observe that fine-tuning for as few as 50k steps is often sufficient. By contrast, training a new model from scratch is more than 12 times as expensive. Table 3 provides a detailed comparison between data mixing and fine-tuning for similar mixing ratios.

To conclude, both fine-tuning models pretrained on **globe-tl** as well as choosing an appropriate data mix during training can be viable approaches to navigate the trade-off between cultural diversity and optimizing performance on Western-oriented, but well-studied benchmarks, such as ImageNet.

### 3.5 Quality Filters

We apply quality filtering using an internal model trained on image-text pairs to calculate the image-text similarity score, in order to assess if our empirical findings continue to hold in this context. The model was trained on global data to make sure it accurately assesses quality for global data, of which English data is a substantial fraction, and its threshold was tuned to balance quality and quantity. We filter out about 60% of the data in our experiments. Appendix B shows that our main

findings continue to hold even when quality filters are applied. For instance, quality-filtered **globe-tl** performs better than quality-filtered **en** on 0-shot Dollar Street, GLDv2, and GeoDE but performs worse on Western-oriented benchmarks, such as ImageNet and COCO retrieval. In addition, the improvement in quality-filtered **globe-tl** over quality-filtered **en** is particularly significant for few-shot geo-localization tasks.

## 4    Related Work

A range of prior work has studied biases in zero-shot classifiers related to a range of sensitive attributes including gender, race and age [1, 25, 27, 12, 24] and found that CLIP models perpetuate biases present in the training data [7, 23, 2]. More recent work extends this analysis to zero-shot performance across groups of different income levels and geographic regions [43, 78]. While several of these papers highlight the central role of training data and the potentially large impact of design choices such as the data source or filtering techniques [22] or translation of English captions for cross-modal multilingual encoders [10, 11, 45], to our knowledge, we are the first to study the impact of image–text pair filtering and text translation on cultural understanding in contrastive VLMs.

Contrastive models are usually evaluated on a range of benchmark datasets including ImageNet [50] (and variations [29, 5]), COCO [38] and Flickr30K [71]. These datasets have been shown to reflect a heavy Western bias [53, 16, 55]. Over the last few years, a range of alternative benchmarks have been proposed, including DollarStreet [49], GeoDE [47], Crossmodal-3600 [58], GLDv2 [66], GeoNet [33], Geo-YFCC [19], xGQA [44], MaXM [13], Ego4D [26] and GD-VCR [70]. While we have used some of these in our experiments, we decided against using others for the following reasons. GeoNet uses only images from North America and Asia and is hence not sufficiently diverse. GeoYFCC contains images from 62 different countries, but Europe-centric and based on images with noisy tags [47]. xGQA mainly evaluates multilinguality: the starting point for its creation is a monolingual English dataset that was then translated. MaXM is an adaptation of XM3600 (which we already use) to multilingual VQA, which contrastive VLMs do not support. Ego4D contains images from only 9 countries with a majority of the images from English-speaking countries (US, UK). GD-VCR is a geo-diverse commonsense benchmark, but is a VQA task unsuited for contrastive VLMs.

The closest work to ours is [48], which argues that progress in global data (Dollar Street and GeoDE) has been much slower than the progress on ImageNet. Unlike their work, however, we study the impact of the training data mixture, study the impact of translation, suggest a setup where both types of metrics can be improved, and propose geo-localization as an evaluation metric. We also consider a broader set of datasets in our study, such as XM3600, GLDv2, and MaRVL.

## 5    Limitations and Future Work

While our work highlights the importance of incorporating cultural and socioeconomic diversity considerations into contrastive VLMs, several limitations should be acknowledged. Firstly, our experimental results are based on recently popular contrastive, encoder-only SigLIP models. While our analysis offers valuable insights, it should be extended to generative VLMs [40, 37, 77, 3, 14, 59, 61, 63, 4, 68, 65, 64, 62]. Secondly, our work primarily highlights the importance of utilizing all available data when pretraining foundation models, but there is potential for additional measures to further improve cultural diversity, such as via regularization, data balancing, or weight averaging [30].

Table 3: A comparison between data mixing and fine-tuning using identical ratios of **en** vs. **globe-tl** examples; e.g., **en** 5 : 1 **globe-tl** for **en** → **globe-tl** means that the model was pretrained on **en** data for 508k steps and then fine-tuned on **globe-tl** data for approximately 102k steps.

| Proportions | 0-shot accuracy on GLDv2 | | | 0-shot accuracy on ImageNet | | |
|---|---|---|---|---|---|---|
| | Data mixing | en → globe-tl | globe-tl → en | Data mixing | en → globe-tl | globe-tl → en |
| **en** 5 : 1 **globe-tl** | 46.43% | 46.24% | 44.81% | 70.28% | 68.45% | 70.87% |
| **en** 2 : 1 **globe-tl** | 46.04% | 49.22% | 44.29% | 70.18% | 68.36% | 70.43% |
| **en** 1 : 1 **globe-tl** | 48.31% | 48.25% | 45.27% | 69.67% | 68.12% | 70.65% |
| **en** 1 : 2 **globe-tl** | 48.70% | 49.22% | 45.33% | 69.02% | 68.18% | 70.46% |
| **en** 1 : 5 **globe-tl** | 49.68% | 49.81% | 45.91% | 68.92% | 67.92% | 70.33% |

Thirdly, although our study consciously separates cultural diversity and multilinguality, investigating their intersection presents an intriguing subject for future research. Fourthly, acknowledging the vagueness of the notion of culture and cultural diversity, we recognize that our experiments are mainly comparing model performance across different countries, regions, or income groups. We do not offer a precise definition of cultural diversity in the context of VLMs and do not claim to cover all aspects of cultures in our analysis. Lastly, we do not address the relationship between cultural diversity and social biases that have been shown to be perpetuated by VLMs. Exploring these connections presents an opportunity to develop more inclusive AI systems.

## 6 Conclusion

This work highlights the importance of considering cultural diversity when training contrastive VLMs. We recommend that researchers and practitioners move away from training models on English-only image–text pairs. While this approach may seem beneficial when considering performance on popular benchmarks, such as ImageNet and COCO, it discards a vast amount of valuable and culturally diverse training information and disproportionately hurts communities of lower socioeconomic status. Our findings suggest that (i) pretraining on the full dataset followed by *short* fine-tuning on English-only data, or (ii) pretraining on a mixture of data, achieve good performance on standard benchmarks while also promoting cultural awareness. When doing so, it is important to acknowledge that there seems to be a trade-off between optimized performance on standard benchmarks and maintaining cultural diversity. Practitioners should, hence, carefully consider the intended use case and the importance of cultural understanding of the resulting model when deciding their pretraining data mixture, also taking into account unintended biases possibly manifesting in downstream models and applications. In specific scenarios, where downstream use is limited to English and multilingualism is not required, we have found that translating the training data into English is a viable option. However, the latter approach should be applied judiciously to avoid sacrificing valuable cultural context within the data.

## Acknowledgement

The authors would like to thank Michael Tschannen and Jeremiah Harmsen from Google DeepMind for their feedback on earlier drafts of this manuscript, as well as Tobias Weyand and Maribeth Rauh from Google DeepMind for the helpful discussions.

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

# A  Appendix

## A.1  Model Card

Model details following Mitchell et al. [42].

- **Model Architecture**: The model architecture contains two towers, a vision transformer encoder [18] and a language transformer encoder [60], both of size B. Models are trained using a contrastive pretraining technique with sigmoid loss [75].

- **Inputs**: The vision encoder takes an image reshaped to $256 \times 256$ as input. The text encoder takes tokenized text cropped to the first $64$ tokens as input.

- **Outputs**: The vision and text encoders both output a $d$-dimensional feature vector where $d = 768$.

- **Intended Use**: The primary use is to conduct research on multimodal applications, such as zero-shot classification and retrieval. We use the models to study the impact of training data filtering on cultural diversity.

- **Known Caveats**: As noted in several prior works, multimodal systems can pick up societal biases. While we demonstrate some of those issues in this work, our analysis is necessarily limited in scope.

- **System Description**: Models are analyzed in a stand-alone setting and not used as part of a larger system.

- **Upstream Dependencies**: None

- **Downstream Dependencies**: None

- **Hardware & Software**: Models are developed using JAX [9] and Flax [28] in the Big Vision [6] codebase. They are trained on Google Cloud TPUs.

- **Compute Requirements**: Each model is trained on $16 \times 16$ TPU chips on 10B seen image-text pairs. A typical training run takes 3.3 days.

- **Model Initialization**: The model is trained from a random initialization.

- **Model Size**: Each SigLIP model has a ViT B/16 image encoder and a size B text encoder.

- **Training Dataset**: We use different subsets of WebLI [15], which consists of images with alt-texts from the public web.

- **Evaluation Datasets**: We evaluate the models on ImageNet-ILSRCV2012 [17], MS COCO [38], Dollar Street [49], Google Landmarks Dataset v2 [66], GeoDE [47], MaRVL [39] and Crossmodal-3600 [58].

# B  Impact of Quality Filters

To assess the impact of quality filters on our findings, we train two models, **en** and **globe-tl**, on 1B image-text pairs, following the same training setup used in the paper. Table 4 shows a summary of these results. We observe that our empirical findings also hold in this setting. For instance, quality-filtered **globe-tl** performs better than quality-filtered **en** on 0-shot Dollar Street, GLDv2, and GeoDE but performs worse on Western-oriented benchmarks, such as ImageNet and COCO retrieval. In addition, the improvement in few-shot geo-localization for quality-filtered **globe-tl** over quality-filtered **en** is particularly significant.

Table 4: Applying quality filters to SigLIPs does not change the primary conclusions. Filtering training data to English image-text pairs continues to negatively impact cultural diversity even though it improves performance on standard benchmarks.

| | en | globe-tl | en vs. globe-tl |
|---|---|---|---|
| **Culturally diverse zero-shot evaluations** | | | |
| Dollar Street | 46.05% | 48.28% | +2.23% |
| GLDv2 | 28.21% | 30.67% | +2.46% |
| GeoDE | 90.53% | 90.57% | +0.04% |
| **Prevalent Western-oriented benchmarks** | | | |
| 0-shot ImageNet | 66.96% | 66.32% | −0.64% |
| COCO I→T R@1 | 56.72% | 52.80% | −3.92% |
| COCO T→I R@1 | 37.33% | 34.46% | −2.87% |
| **10-shot geo-localization** | | | |
| Dollar Street (country) | 9.40% | 9.82% | +0.42% |
| GeoDE (country) | 10.10% | 14.73% | +4.63% |
| GeoDE (region) | 21.97% | 28.22% | +6.25% |

