# OpenReview forum: "No Filter: Cultural and Socioeconomic Diversity in Contrastive Vision-Language Models"
_NeurIPS.cc/2024/Conference — NeurIPS 2024 poster_

### Official Review · Reviewer_nvDK · 2024-06-19

**Soundness:** 2
**Presentation:** 3
**Contribution:** 1
**Rating:** 5
**Confidence:** 3

**Summary:**

This paper discusses the impact of the common practice of filtering for English-only data on training vision-language models, showing that this negatively impacts performance on tasks covering diverse cultural regions and backgrounds. The work proposes foregoing this filtering step to improve this representation, as well as potentially using a short English-only fine-tuning stage to achieve a more acceptable balance between standard task performance and diversity.

**Strengths:**

Bias in vision-language models, and generally in machine learning, is an important problem. The proposed mitigation method (training on global data without English-only filtering) makes sense and the results seem to support this practice.

**Weaknesses:**

The contribution of the paper seems limited, as it is unsurprising that training on global data improves performance on benchmarks measuring global data understanding. The findings are not contextualized with prior works on the effect of training CLIP-style models on large, weakly-curated datasets or for multilingual understanding [1-3]. Results are only shown for one particular type of dual-encoder VLM (SigLIP). There is also no consideration of the effect of limited resources for training, which may make training on a larger less-curated dataset less feasible and use more computational resources, which may disproportionately impact global research communities, calling into question the paper’s categorical call to stop filtering such datasets for English-language data (L161).


[1] Carlsson et al. Cross-lingual and multilingual CLIP. LREC 2022

[2] Chen et al. mclip: Multilingual clip via cross-lingual transfer. ACL 2023

[3] Cherti et al. Reproducible scaling laws for contrastive language-image learning. CVPR 2023

**Questions:**

Why is only SigLIP tested? Even restricted to CLIP-style models, it seems like the same methodology could be applied to any vision-language dual-encoder model, and to initialization schemes other than the random initialization used (L577) which presumably could have a significant effect on global representation.

Can you elaborate on how filtering hurts performance for “low-income households” (L50)? The datasets used (L98 on) are explicitly chosen to cover diverse geographical regions, which includes developing countries, but it is not clear whether socioeconomic status has a clear effect on results when controlling for geographic region and/or language. This also relates to the paper title which explicitly mentions socioeconomic diversity, though the paper seems to demonstrate the effect of geographic diversity specifically.

**Limitations:**

Overall the presentation of limitations is thoughtful and fairly exhaustive. There lacks a discussion of resource and compute limitations, which affect the ability to train on larger, less-filtered datasets.

---

> ### Author Rebuttal · Authors · 2024-08-06
>
> We would like to thank the reviewer for the careful and insightful feedback. We especially appreciate their positive assessment of the importance and presentation of our work. We provide our answers to the reviewer’s raised questions and concerns below, are looking forward to their response, and hope for a favorable reassessment of our scores.
>
> **Q1: Shouldn’t it be expected that the *globe* models outperform the *en* model on benchmarks measuring global understanding?**
>
> While we agree that this might seem intuitive at first sight, it is not necessarily true. There are English websites published in most countries and pictures of landmarks are frequently captured by tourists, so it is possible that the English subset of the web is sufficient to capture cultural diversity. Our contribution is to provide a comprehensive evaluation of cultural diversity in SOTA contrastive VLMs, and we hope that these results suffice to clear any lingering doubts about the necessity of training on global data when building foundation models. Please refer to our global response for more detailed information.
>
> **Q2: Are the findings related to prior work on multilingual understanding?**
>
> We thank the reviewer for the additional pointers to prior work on multilingual understanding and will make sure to include these references in the revised version of our paper. In the context of this work, however, we purposely disentangled multilinguality from cultural diversity because we believe these measure different aspects. As noted in Section 5, we do, however, see the investigation of their intersection as an intriguing subject for future research and are looking forward to new insights in that area!
>
> **Q3: Do the findings hold for other contrastive VLMs beyond SigLIP (e.g., CLIP)?**
>
> SigLIP and CLIP operate on the same principle of aligning representations/embeddings for texts and images and the difference is only in the choice of the loss function. We expect the same results to hold for CLIP as well. The reason we use SigLIP is because of its superior performance and widespread use (>1M downloads/month in Huggingface).
>
>
>
> **Q4: How does additional, quality-based data filtering affect the findings presented in this work?**
>
> Thank you for raising this question. We have conducted new experiments to verify this. Please refer to the global response for a summary of our findings. Generally, we observe that the same conclusions hold even after applying quality filters based on image-text similarity. We will add more details about this in the revised version of the paper.
>
> **Q5: Are the findings limited to geographic diversity?**
>
> While geographic and socioeconomic diversity are inherently linked due to different income levels and economic circumstances in countries and regions around the world, Dollar Street does contain data from different income groups for every region. Kindly refer to Figure 3 (left), in which we specifically address these differences.
>
> **Q6: Does the proposed removal of the English-only data filter increase computational cost?**
>
> We believe our findings are orthogonal to having additional, quality-based data filters that can be applied in practice to limit computational cost incurred during training. In particular, while we do not challenge the use of quality filters, our work warns against using those filters that are based on English-only datasets or favor English-only captions. Please refer to the global response for a more detailed response.

---

> > ### Author Response · Authors · 2024-08-11
> > **Follow up**
> >
> > Dear reviewer,
> >
> > Thank you again for the detailed and constructive comments.
> >
> > As the discussion period is about to end, we would like to ensure we've addressed all of your questions and concerns. If you feel we have satisfactorily responded, please let us know. Otherwise, please let us know your remaining concerns so we can address them before the discussion period closes.
> >
> > Thank you

---

> > ### Comment · Reviewer_nvDK · 2024-08-11
> >
> > Thank you for your thoughtful response. My concerns are generally addressed, particularly the point that all experiments are compute-matched. I would recommend highlighting this in the revised version and acknowledging that underrepresented communities may have limited compute resources. I am still not fully convinced that the result is unexpected (it seems natural that landmarks are more captured in global data) but the findings seem overall valuable to the community and I have adjusted my rating accordingly.

---

> > > ### Author Response · Authors · 2024-08-11
> > >
> > > Dear reviewer,
> > >
> > > Thank you for engaging with us and for adjusting your rating. We appreciate the detailed and constructive comments, and we are happy that we have addressed your concerns. We will make sure to highlight these points in future revisions of our work.
> > >
> > > Sincerely

---

### Official Review · Reviewer_JWWG · 2024-07-12

**Soundness:** 3
**Presentation:** 2
**Contribution:** 3
**Rating:** 6
**Confidence:** 3

**Summary:**

The paper shows that vision-language models trained solely on English filtered data displays a bias towards western-centric benchmarks. They then present a simple solution for training models that perform well on globally diverse datasets while not sacrificing performance on gold standard datasets such as ImageNet. They accomplish this by training on the unfiltered dataset then fine-tuning on the relevant subset for the downstream task.

**Strengths:**

**Originality:**

The paper presents a relevant and unaddressed problem of cultural bias. The remedy of only filtering for english data after pretraining is simple and effective.

**Quality:**

The experiments are thorough and properly support the claims of the paper.

**Significance:**

The paper could impact how researchers train foundational models and help to mitigate cultural bias in future models.

**Clarity:**
The paper is reasonably clear to understand. There are small issues such as with figure 1, but nothing glaring.Overall the weaknesses are minor.

**Weaknesses:**

Overall the weaknesses are minor.

- One potential confounding factor is CLIP-filtering which most text-image models use to improve accuracy. It’s unclear how this approach would interact with this and other filtering techniques. For example, it’s known that CLIP models [1] tend to filter out non-english captions.

- One other potential weakness is that the technique could be computationally expensive since CLIP needs to be retrained on a potentially large English subset.

**Questions:**

Minor Comments:
- It would be useful to add the size of WebLI for comparison to LAION since most CLIP-like models have been trained on LAION.
- I think the commas in the paper have been messed up for the numbers.



I think this statement in the contributions is a bit strong and overclaiming. The logical leap from zero-shot classification on Dollar Street to harming low socioeconomic groups is a bit far:
"filtering the training data to English image–text pairs negatively impacts cultural diversity and disproportionately hurts communities of lower socioeconomic status, exacerbating existing disparities. Its impact is demonstrably captured by zero-shot classification accuracy on Dollar Street."

**Limitations:**

Limitations are adequately addressed.

---

> ### Author Rebuttal · Authors · 2024-08-06
>
> We would like to thank the reviewer for the careful and insightful feedback. We especially appreciate their positive assessment of the originality, quality, significance, and clarity of our work. We hope that our rebuttal below addresses all of the reviewer’s questions, are happy to provide more details, and look forward to the reviewer’s response to our rebuttal.
>
> **Q1: How does additional, quality-based data filtering affect the findings presented in this work?**
>
> Thank you for raising this question. We have conducted new experiments to verify this. Please refer to the global response for a summary of our findings. Generally, we observe that the same conclusions hold even after applying quality filters based on image-text similarity. We will add more details about this in the revised version of the paper.
>
> **Q2: Does the proposed removal of the English-only data filter increase computational cost?**
>
> We believe our findings are orthogonal to having additional, quality-based data filters that can be applied in practice to limit computational cost incurred during training. In particular, while we do not challenge the use of quality filters, our work warns against using filters that are based on English-only datasets or favor English-only captions. Please refer to the global response for a more detailed response.
>
> **Q3: Size of WebLI**
>
> As is detailed in Section 2 of our paper, all models are trained on 10B image–text pairs (roughly 610k training steps). Hence, all models are compared on a *compute-matched* setup: *en* models are trained for ~3 epochs, while *globe* models are trained for a single epoch. Please refer to the global response for more details.
>
> **Q4: What evidence supports the claim that English-only data filtering exacerbates existing disparities?**
>
> We show examples of this in Figure 1.c and Figure 3 (left). For instance, accuracy on Dollar Street for low-income groups ($0-200) drops from 35% using *globe-tl* to less than 29.9% when trained on English-only data. Similarly, accuracy on examples from the African continent drops from 38.4% using *globe-tl* to less than 35.8% using English-only data. Because of this, pretraining on English-only exacerbates existing disparities because there is already a large gap in performance between low-income and high-income regions (as shown in the figures) and training on English-only data increases those gaps.
>
> We also thank the reviewer for the comment regarding the commas and will make sure to carefully check this for future revisions of our paper.

---

> > ### Comment · Reviewer_JWWG · 2024-08-08
> >
> > I thank the authors for their response. My concerns have been addressed in the rebuttal. I am maintaining my rating of weak accept. My justification is that the paper provides some good empirical insights on how to handle english and non-english data for large-scale VLM training. I think the scope and generality of the findings are relatively small which is why my score isn't higher, but I like the paper and think it could be useful for other researchers.

---

> > > ### Author Response · Authors · 2024-08-11
> > > **Follow up**
> > >
> > > Dear reviewer,
> > >
> > > Thank you for engaging with us. We appreciate the detailed and constructive comments, and we are happy that we have addressed your concerns.
> > >
> > > Sincerely

---

### Official Review · Reviewer_mxHk · 2024-07-13

**Soundness:** 3
**Presentation:** 3
**Contribution:** 2
**Rating:** 5
**Confidence:** 3

**Summary:**

For the field of training contrastively learned-based VLM, this paper firstly displays that training from English data would lead to worse cultural diversity in zero-classification evaluation. By discarding the influence of languages in evaluation, this paper proposed a geo-localization task, which could observe a trained model underperforms globe and globe-tl data. Besides, the paper also explores the possible strategies for achieving a better trade-off between standard performance and cultural diversity. According to the observations, the authors found that pre-train on the globe and fine-tuning in English is the optimal strategy with less cost.

**Strengths:**

- The paper points out that cultural diversity is important but neglected in standard benchmarks.
- The paper discussed the practice of development strategies in pretrain and fine-tuning the constrastive VLM for the trade-off between standard performance and cultural diversity, which makes the practice benefit from this research.

**Weaknesses:**

- The writing is not clear enough. The structure and relation between different parts of this paper is not clear.
- The necessity of the proposed geo-localization task remains unclear in this paper. We don’t know the necessity of geo-localization compared with prior metrics that could reflect the cultural diversity of constrastive VLM.

**Questions:**

- Could you please restate the definition of the concept of Socioeconomic Diversity for me? Additionally, how can it be measured? What level of diversity do we expect the model to achieve?

**Limitations:**

Yes

---

> ### Author Rebuttal · Authors · 2024-08-06
>
> We would like to thank the reviewer for the careful and insightful feedback. We provide our answers to the reviewer’s raised questions and concerns below, are looking forward to their response, and hope for a favorable reassessment of our scores.
>
> **Q1: How is the paper structured?**
>
> We are sorry to hear that the reviewer found the structure of our paper to be unclear. We motivate and present a brief overview of our work in Section 1, before moving on to discussing the chosen model architecture as well as training and evaluation datasets in Section 2. We also briefly summarize our key findings at the end of Section 2 and explain them in more detail in Section 3. We conclude with a discussion of related work, limitations and future work. If the reviewer has any specific suggestions for improving clarity, we would be happy to hear them and incorporate them in the revised version of the paper.
>
> **Q2: How does the proposed geo-localization task compare to alternative metrics of cultural diversity in contrastive VLMs?**
>
> As can be seen in Figure 5 (right), the proposed geo-localization task is well correlated with other metrics of cultural diversity in contrastive VLMs (such as zero-shot classification on Dollar Street, GLDv2, GeoDE and MaRVL). One important distinction, however, is that the few-shot geo-localization task only evaluates the learned image embeddings while discarding the text tower, which can provide insightful additional information. In addition, it provides a very strong signal as shown in Table 2, where differences in accuracy can reach up to 10%.
>
> **Q3: How is socioeconomic diversity defined and measured in the context of this work?**
>
> As noted in Section 5, we do not claim to offer a precise definition of cultural or socioeconomic diversity in the context of VLMs. Instead, we use model performance across different countries, regions, and income groups as a useful proxy for these concepts. We acknowledge in Section 5 (limitations) that this may not cover all aspects of cultural and socioeconomic differences.

---

> > ### Author Response · Authors · 2024-08-11
> > **Follow-up**
> >
> > Dear reviewer,
> >
> > Thank you again for the detailed and constructive comments.
> >
> > As the discussion period is about to end, we would like to ensure we've addressed all of your questions and concerns. If you feel we have satisfactorily responded, please let us know. If you have any further questions, we are happy to address them before the discussion period closes.
> >
> > Thank you

---

> > ### Comment · Reviewer_mxHk · 2024-08-12
> > **Official Comment by Reviewer mxHk**
> >
> > Thanks for the response which addressed most my concerns. I would keep my rating.

---

### Official Review · Reviewer_dZWv · 2024-07-14

**Soundness:** 2
**Presentation:** 3
**Contribution:** 3
**Rating:** 6
**Confidence:** 4

**Summary:**

The paper compares performance of VLMs trained on just English data, multilingual data and the English translations of the multilingual data, on a range of benchmarks including Western-centric ones as well tasks that involve geographically diverse inputs. The paper finds that (i) training on just English data negatively impacts cultural diversity, (ii) existing multilingual evaluations may not work as well as intended due to their images still being Western-centric, (iii) it is possible to improve cultural understanding without sacrificing performance on the popular Western-centric benchmarks

**Strengths:**

1. The evaluations are done throughly, and a new test is proposed, which addresses limitations of existing evaluations (e.g. XM3600 images are still Western-centric).

2. Despite some concerns about the setup (see Weaknesses below), I think the paper raises an important message and some of the findings probably would hold for other training setups

3. The paper proposes a new training pipeline to improve cultural understanding without impacting performance on Western benchmarks (ImageNet). This probably has practical value for a lot of ML practitioners.

**Weaknesses:**

1. In the "Summary of Findings" section, the claims should be made more specific and grounded in evidence, e.g.

(i) "exacerbating existing disparities" --> the authors should show how much the performance *gap* between best and worst groups increase by training on **en** versus **globe**,

(ii) instead of saying "world knowledgeable", quote specific numbers and findings.

2. In figure 1 (and other parts of the paper), the authors claim that models trained on only English data do poorly on geographically benchmarks, e.g. by confusing landmarks with similar ones in Western countries. However, this result is expected given that the **en** data is a subset of **globe**, and the landmarks test may be in-distribution for models trained on **globe** and out-of-distribution for those trained on **en**. I think the paper's claim could be made stronger by better controlling for these confounding factors (e.g. by removing test-set knowledge contamination).

3. Besides, I find the dataset construction used in the experiments unrealistic: the authors state that "globe denotes the raw, multilingual data with minimal filtering applied" - many VLMs are trained on highly filtered multilingual datasets (e.g. LAION)

**Questions:**

1. Since **en** is a subset of **globe** (and thus assumably containing fewer training samples), are all models just trained until convergence? How are the different training set sizes controlled for in the experiments?

2. In Conclusion, the authors claim that "there seems to be a trade-off between optimized performance on standard benchmarks and maintaining cultural diversity" - I thought the paper already shows in Section 3.4 that this trade-off could be avoided?

**Limitations:**

The authors have sufficiently address the limitations.

---

> ### Author Rebuttal · Authors · 2024-08-06
>
> We would like to thank the reviewer for the careful and insightful feedback. We especially appreciate their positive assessment of the thoroughness, importance and practicality of our work. We hope that our rebuttal below addresses all of the reviewer’s questions, are happy to provide more details, and look forward to the reviewer’s response to our rebuttal.
>
> **Q1: How much does training on en instead of globe increase the performance gap between groups of high and low socioeconomic status?**
>
> We show examples of this in Figure 1.c and Figure 3 (left). For instance, zero-shot classification accuracy on Dollar Street for low-income groups ($0-200) drops from 35% with *globe-tl* to less than 29.9% with *en*. Similarly, accuracy on examples from the African continent drops from 38.4% using *globe-tl* to less than 35.8% using *en*. Because of this, pretraining on English-only data exacerbates existing disparities because there is already a large gap in performance between low-income and high-income groups (as shown in the figures) and training on English-only data increases these performance gaps.
>
> **Q2: How does additional, quality-based data filtering affect the findings presented in this work?**
>
> Thank you for raising this question. We have conducted new experiments to verify this. Please refer to the global response for a summary of our findings. Generally, we observe that the same conclusions hold even after applying quality filters based on image-text similarity. We will add more details about this in the revised version of the paper.
>
> **Q3: Shouldn’t it be expected that the globe models outperform the en model on benchmarks measuring global understanding?**
>
> Not necessarily. Please refer to the global response for our answer to this question.
>
> **Q4: How and for how long are the models trained?**
>
> As is detailed in Section 2 of our paper, all models are trained on 10B image-text pairs (roughly 610k training steps). Hence, all models are compared on a *compute-matched* setup: *en* models are trained for ~3 epochs, while *globe* models are trained for a single epoch. Please refer to the global response for more details.
>
> **Q5: Can the trade-off between good performance on standard benchmarks and maintaining cultural diversity be avoided?**
>
> While pre-training on global data and fine-tuning on English data allows to balance performance on standard and culturally diverse evaluation metrics, neither fine-tuning nor data mixing allow for complete avoidance of the tradeoff between these metrics (as can be seen in Figure 5 (left)).  It is however possible that other approaches (such as different training paradigms or model weight merging) might lead to improved tradeoffs and maybe eventually even a complete avoidance thereof. We are looking forward to future work in this area!

---

> > ### Author Response · Authors · 2024-08-11
> > **Follow-up**
> >
> > Dear reviewer,
> >
> > Thank you again for the detailed and constructive comments.
> >
> > As the discussion period is about to end, we would like to ensure we've addressed all of your questions and concerns. If you feel we have satisfactorily responded, please let us know. If you have any further questions, we are happy to address them before the discussion period closes.
> >
> > Thank you

---

> > > ### Comment · Reviewer_dZWv · 2024-08-12
> > > **Response to rebuttal**
> > >
> > > Thank you for the rebuttal and the new results. The authors' response still doesn't completely resolve my concerns with regards to the following points:
> > >
> > > Q1: The "exacerbating existing disparities" claim - I suggested backing up this claim by looking into the performance *difference* between best and worst groups, from training on en versus globe, but the authors only reported the worst-group performance. As mentioned earlier, overall the claims in "Summary of Findings" should also be more appropriately phrased and corroborated.
> > >
> > > Q2: How was the "quality score filter based on image-text similarity" implemented? Was it tuned separately for en and globe-tl? More details would be helpful.

---

> > > > ### Author Response · Authors · 2024-08-13
> > > > **Thank you**
> > > >
> > > > Dear reviewer,
> > > >
> > > > Thank you for engaging with us. Please see our response below:
> > > >
> > > > **Q1.** We apologize for not responding clearly to this point. We do agree that the claim of "exacerbating existing disparities" is supported empirically by looking into the performance difference between best and worst groups. That was actually what we meant to refer to in our earlier response. We will make this more explicit in the revised version of the paper. In dollar street, the performance gap between low-income groups (0-200) and high-income groups (>1998) is 32.5% using English-only data. The gap shrinks to 27.4% using *globe-tl*. We will add the full table below, of accuracy disaggregated by income leel, to the supplementary materials.
> > > >
> > > > | Data | | 0-200 (worst) | | 201-684 | | 685-1997 | | >1998 (best) | | Performance Gap |
> > > > | ---- | - |------------ | - | --------- | - | ---------- | - | ------ | - | --------------- |
> > > > | en | | 29.9 | | 44.4 | | 55.5 | | 62.4 | | 32.5 |
> > > > | globe | | 32.9 | | 45.1 | | 56.3 | | 62.3 | | 29.4 |
> > > > | globe-tl | | 35.0 | | 48.1 | | 56.9 | | 62.4 | | 27.4 |
> > > >
> > > > **Q2.** For quality filtering, we used an internal CLIP-like model that was previously trained on image-text pairs to calculate the image-text similarity score. The model was trained on global data to make sure it has a good assessment of quality for global data, of which English is a substantial fraction, and its threshold was tuned to balance between quality and quantity. We filter out about 60% of the data in our experiments.
> > > >
> > > > We hope these answer your questions, and resolve your remaining concerns.

---

> > > > > ### Comment · Reviewer_dZWv · 2024-08-13
> > > > > **Response**
> > > > >
> > > > > Thank you for the additional clarification! I support the acceptance of the paper as it helps bring attention to the biases and implications of training on English-only data. But I would keep my original rating of "Technically solid, moderate-to-high impact paper, with no major concerns" rating given my overall assessment of the contributions.

---

### Author Rebuttal · Authors · 2024-08-06

We would like to thank all reviewers for the detailed and constructive questions and comments. We especially appreciate the positive feedback on the thoroughness of our experiments, significance of findings, and clarity of work. In the below, we answer the concerns shared by multiple reviewers. We hope that these, in addition to our responses to individual reviewers, help clarify any open questions.

**Q1: How does additional, quality-based data filtering affect the findings presented in this work?**

This is an excellent point and we plan to include this in the revised version of the paper. We have pretrained two additional SigLIP models with quality-based filtering on 1B image–text pairs each: *en* and *globe-tl*. We evaluate these on Dollar Street, GeoDE, ImageNet and COCO. We observe that the same qualitative conclusions continue to hold even after applying a quality filter based on image-text similarity. For instance, *globe-tl* performs better than *en* on 0-shot Dollar Street, GLDv2, and GeoDE but performs worse on Western-oriented benchmarks, such as ImageNet and COCO retrieval. In addition, the improvement in geolocalization is particularly significant.

| Task | en | globe-tl |
| ---- | -- | -------- |
|Dollar Street | 46.1% | 48.3% |
|GLDv2 | 20.7% | 28.2% |
|GeoDE | 90.5% | 90.5% |
|0-shot ImageNet | 67.0% | 66.3% |
|COCO I2T R@1 | 56.7% | 52.8% |
|COCO T2I R@1 | 37.3% | 34.5% |
|Dollar Street 10-shot | 9.4% | 9.8% |
|GeoDE 10-shot (country) | 10.1% | 14.7% |
|GeoDE 10-shot (region) | 22.0% | 28.2% |

We plan to additionally train all three variants *en*, *globe*, and *globe-tl* for 10B quality-filtered image–text pairs as was done in the paper, and include the results in the supplementary material of the paper. But as shown above, the same conclusions hold even after applying quality filters.





**Q2: Shouldn’t it be expected that the globe models outperform the en model on benchmarks measuring global understanding?**

We thank the reviewers that have raised this important question. While this might seem intuitive at first sight, it is not necessarily true. There are English websites published in most countries and pictures of landmarks are frequently captured by tourists, so it is possible that the English subset of the web is sufficient to capture global diversity. In addition, we have found that certain seemingly global datasets (such as XM3600) seem to fail in capturing cultural nuances. Hence, a key contribution of our work is identifying four zero-shot classification datasets actually capturing global understanding, as well as introducing the few-shot geo-localization task. Given that this is the first comprehensive evaluation of cultural diversity in SOTA contrastive VLMs as far as we know, we hope that these results suffice to clear any lingering doubts about the necessity of training on global data when building foundation models.

**Q3: How and for how long are the models trained?**

As is detailed in Section 2 of our paper, all models are trained on 10B image–text pairs (roughly 610k training steps). Hence, all models are compared on a *compute-matched* setup: *en* models are trained for ~3 epochs, while *globe* models are trained for a single epoch. We have observed that performance gaps between *en* and *globe* models continue to persist even for significantly longer training durations. We will add more details about this last point in the revised version of the paper.

**Q4: Does the proposed removal of the English-only data filter increase computational cost?**

All models presented in our paper incur the same computational cost due to being trained for the same number of total examples seen (albeit different numbers of unique examples). As can be seen in our response to Q1, our findings are orthogonal to having additional, quality-based data filters that are often applied in practice to limit computational cost incurred during training. In particular, while we do not challenge the use of quality filters, our work warns against using filters that are based on English-only datasets or favor English-only captions.


We are grateful again for the reviewers' detailed and constructive feedback. If we have satisfactorily answered your questions, we hope you would consider revising your scores. Otherwise, please let us know if there are any other questions or concerns, so we can respond to them during the discussion period.

---

### Decision · Program_Chairs · 2024-09-25

**Decision:**

Accept (poster)

**Comment:**

The paper presents a timely and valuable contribution to addressing cultural bias in vision-language models (VLMs), demonstrating that training on global data improves cultural understanding—without significantly sacrificing performance on standard benchmarks. All the reviewers comments have been addressed. All the scores recommend acceptance. Congrats to the authors.

Pre-training on global data and fine-tuning with English provides a practical and simple solution for balancing performance across culturally diverse datasets. The paper introduces new evaluation tasks, such as geo-localization, which strengthen its contributions.

While some concerns remain regarding the novelty of certain findings and the clarity of the structure, the reviewers generally agree that the work raises important issues on dataset biases and will benefit the broader machine learning community. The paper’s thorough evaluations and promising results support its relevance and potential impact.

Despite minor reservations about computational costs and dataset filtering, these concerns are orthogonal to the current contribution and ripe areas for future research.